# High-Frequency Electromagnetic Induction (HFEMI) Sensor Results from IED Constituent Parts

**Benjamin Barrowes [1],\*, Mikheil Prishvin [2], Guy Jutras [3] and Fridon Shubitidze [4]**

[1]  ERDC Cold Regions Research and Engineering Laboratory, 72 Lyme Road, Hanover, NH 03755, USA
[2]  Subsurface Sensing Technologies and Consulting, LLC, Racha St. 48, Gldanis Dasaxleba, Tbilisi 0101 , Georgia; mikheil.prishvin@sstechcon.com
[3]  Geophex, Ltd., 605 Mercury Street, Raleigh, NC 27603, USA; jutras@geophex.com
[4]  Thayer School of Engineering, Dartmouth College, Hanover, NH 03755, USA; Fridon.Shubitidze@dartmouth.edu
\*  Correspondence: benjamin.e.barrowes@usace.army.mil; Tel.: +1-603-646-4822

**Abstract:** The detection and classification of subsurface improvised explosive devices (IEDs) remains one of the most pressing military and civilian problems worldwide. These IEDs are often intentionally made with either very small metallic parts or less-conducting parts in order to evade low-frequency electromagnetic induction (EMI) sensors, or metal detectors, which operate at frequencies of 50 kHz or less. Recently, high-frequency electromagnetic induction (HFEMI), which extends the established EMI frequency range above 50 kHz to 20 MHz and bridges the gap between EMI and ground-penetrating radar frequencies, has shown promising results related to detecting and identifying IEDs. In this higher frequency range, less-conductive targets display signature inphase and quadrature responses similar to higher conducting targets in the LFEMI range. IED constituent parts, such as carbon rods, small pressure plates, conductivity voids, low metal content mines, and short wires respond to HFEMI but not to traditional low-frequency EMI (LFEMI). Results from recent testing over mock-ups of less-conductive IEDs or their components show distinctive HFEMI responses, suggesting that this new sensing realm could augment the detection and discrimination capability of established EMI technology. In this paper, we present results of using the HFEMI sensor over IED-like targets at the Fort AP Hill test site. We show that results agree with numerical modeling thus providing motives to incorporate sensing at these frequencies into traditional EMI and/or GPR-based sensors.

**Keywords:** electromagnetic induction; landmine detection; high-frequency EMI; voids; IED

## 1. Introduction

Interrogation of the subsurface using electromagnetics is desirable for the fundamental reason that the ground does not have to be disturbed during interrogation. To acquire information through soils, with typical conductivities in the range of 0.001 to 0.1 Siemens per meter, appropriately low electromagnetic (EM) frequencies must be chosen in order to minimize losses related to the skin effect [1]. Among EM sensing modalities, ground-penetrating radar (GPR), operating from perhaps 50 or 100 MHz to the GHz range, has long been of interest for exploring the subsurface, whether for characterizing the medium or for identifying inclusions. These systems operate by transmitting electromagnetic waves into the ground, sometimes from multiple locations, and then recording reflections over time and space. GPR suffers from the rapid attenuation of its transmitted fields in lossy media, but has the benefit of having a short enough wavelength to guide the radiation in a preferred direction [2–4]. In contrast, electromagnetic induction (EMI) systems operate at much lower frequencies, typically 10 s of Hz up to 50 kHz or 100 kHz. EMI suffers from the inability to direct the radiation due to the long wavelengths and from the low resolution imposed by low frequencies. It is

also limited by a rapid decay in the transmitted fields of $1/r^3$ from the first term of the Hertzian dipole, but has the advantage of being able to treat the soil as if it were transparent in most cases [1,5–9]. In addition, EMI sensing allows the possibility of extracting intrinsic parameters such as the dipole moments from discrete targets that aid in classification. EMI, together with recent inversion models, has performed well, for example, in recent land-based detection and discrimination tests for detecting and classifying metallic targets such as UXO in the near subsurface (i.e., $\approx$1 m) [10–12].

High-frequency electromagnetic induction (HFEMI) operates in between the LFEMI and GPR frequency ranges. Loosely defined as the frequency regime from 100 kHz to 20 MHz, HFEMI produces long enough wavelengths ($\lambda \gg l$) to still satisfy magnetoquasistatic assumptions, i.e., displacement currents are negligible compared to conduction currents ($\frac{\varepsilon\omega}{\sigma} \ll 1$) [1,13–20]. Yet the frequency is high enough that less-conducting, or intermediate electrically conducting materials (IECM), targets respond and produce identifiable signatures. HFEMI retains some traditional EMI practices such as using compact sensors to perform local interrogation, especially for the identification of small scale configurations and discrete inclusions. However, even within this higher frequency realm, the kinds of signals available and the resolution do not readily support the formation of images based on conductivity such as in the case of previous instruments operating in this frequency range such VETEM [21,22], HFS [23,24], and the LASI high-frequency ellipticity system [25] which were generally looking for large and/or deep targets via imaging similar in approach to GPR.

When constructing a broadband EMI sensor, it is usually desirable to avoid coil and reactive resonances. As the wavelength on the excitation and receiving coils shortened as we increased the frequency in our HFEMI sensor, we encountered difficulties with wavelength coherency and noise associated with resonance that we have detailed elsewhere [13,26,27]. At these shorter wavelengths, to maintain field homogeneity over time and space, we had to reduce the number of turns on the Tx and Rx coils (and thereby magnetic moment), thus reducing the SNR of the sensor. As a partial solution, we have investigated hybrid coils using switches that transform from more turns at lower frequencies to fewer turns in parallel at higher frequencies [28], though the results presented here use a fixed number of turns in the coils. In all, we fabricated three types of high-frequency electromagnetic induction sensors using different coil configurations (Geophex method, a centered figure-8 Rx method, and an overlapping "double-O" method) to buck the primary field in the receiver coils (see Figure 1). With these sensors, we have been able to detect emerging carbon fiber shell UneXploded ordnance (UXO) on military ranges [10–12] as well as evidence of depleted uranium both in solid and dispersed forms [29,30].

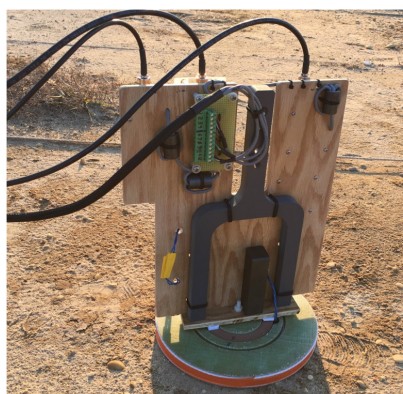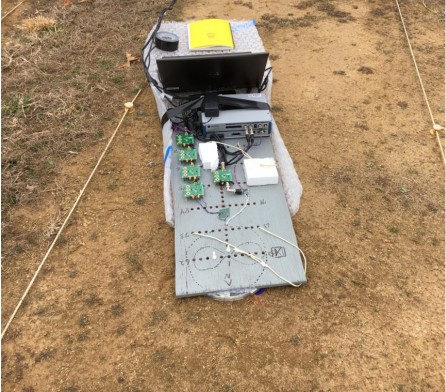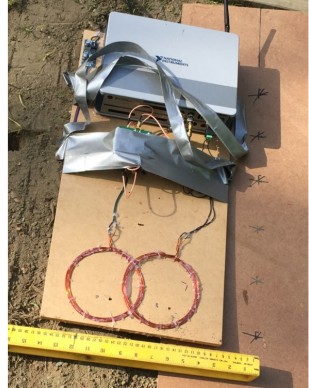

**Figure 1.** Three different variants of the HFEMI sensor: (**Left**) Geophex GEM-3 type with concentric transmitter and receiver coils; (**Center**) figure-8 quadruple receiver type; (**Right**) overlapping coil type.

HFEMI has the potential to contribute to the detection and identification of subsurface IEDs which remain the number one asymmetric threat against the US military worldwide [31–34]. Recent IEDs consist of low or non-metallic constituents and may use intermediate electrical conductivity (IEC) materials ($1 < \sigma < 10^5$ S/m such as Carbon Rods (CR) [35]) and low metal content objects (such as

wires) along with detonator elements and triggering mechanisms. As a result of the conductivity of these IEC materials being less than the conductivity of larger metallic targets such as anti-tank mines, these IEDs are difficult to detect using traditional metal detectors (DC) and EMI instruments (30 Hz to 100 kHz) [13,26,36–39]. Therefore, the detection and identification of CR and wires have become a high priority for the US military during recent conflicts. In addition, because the relaxation spectrum in the inphase and quadrature components of the response from metal targets shifts to higher frequencies the smaller the metal target is, the small metal pins in low metal content landmines are also detectable at HFEMI frequencies [6,31].

The organization of the remainder of this paper is as follows, Section 2 will present electromagnetic induction theory along with special considerations for high-frequency electromagnetic induction. Section 4 will present results from our field test of the HFEMI sensor in all three configurations at the Fort AP Hill IED test lanes. Section 4.3 discusses the physics of wire detection at HFEMI frequencies with accompanying results. This is followed by a conclusion.

## 2. EMI Phenomenology

Electromagnetic induction sensors have been in use for several decades as metal detection instruments. In the case of frequency domain EMI sensors, the alternating primary field is always broadcast from the transmitting coil during data collection, but this complicates the design of the sensor because only the secondary field from the target is desired. For this reason, various schemes of bucking the primary field in the receiver are employed as noted above. In contrast, time domain EMI sensors do not suffer from this problem of trying to subtract out the primary field from the data [1,40–44]. However, to detect eddy currents in objects with lower conductivity than metals, the frequency range of traditional EMI sensors had to be increased. To sense the equivalent of 15 MHz in the time domain, a time domain instrument would need to turn off the primary magnetic field within $\Delta t = \frac{1}{f_{max}}$ (where $\Delta t$ is the transience of the primary field turnoff), or $\Delta t$ = 67 ns, which is currently not feasible [45]. Therefore, we developed a frequency domain instrument that could operate at frequencies up to 15 MHz or more. A more thorough discussion of the iterations of our HFEMI instrument can be found in [26].

The physical principles of frequency domain (FD) EMI are based on inducing eddy currents in conducting targets. According to Lenz's law, a conducting object in a primary alternating magnetic field will develop eddy currents to oppose the changing flux inside the object [46,47] with these eddy currents depending on the conductivity of the object. In turn, these eddy currents produce a secondary alternating magnetic field that is out of phase with the primary field (see Figure 2). In metals, conductivity and electron mobility is high, which allows induced opposing currents to form (again via Lenz's law) at lower frequencies than in lower-conducting materials. In other words, the time rate of change of the inducing primary field needs to be higher (higher frequency) for materials with lower conductivity. We call the component of the eddy current synchronized with the primary field the inphase component and the component that is 90° out of phase with the primary field the quadrature component.

Three salient features in the EMI inphase and quadrature (I and Q) spectrum—a linear range, a quadrature peak, and an inductive limit—occur at different frequency ranges for different objects, but always in the same order [48]. These features can be seen in the results in the following sections. At low frequencies, the eddy currents flow through the bulk of the object [49]. They are 90° out of phase with the primary magnetic field, and have magnitude that grows proportionally to the frequency of the magnetic field. In the low-frequency linear range, material resistivity limits the magnitude of the secondary field. This range is not normally used to characterize UXO targets, but is used for applications such as resistivity estimation of soil. As frequency increases, these eddy currents in the target become larger in magnitude, and a quadrature peak develops in the received magnetic field. The frequency of the quadrature peak is a result of object properties such as size, conductivity, magnetic permeability, and shape. In turn, these properties are linked to signal features central to

inversion and classification. IEC materials the size of most UXO exhibit this peak in the HFEMI band of 100 kHz–20 MHz [13,26,39]. As the frequency of the primary field increases further, the eddy currents move to the surface of the object, and the secondary *B* field plateaus in the inductive limit (i.e., a reflection of the primary field).

Using frequencies above 50 kHz allows HFEMI to obtain EMI spectra for smaller and/or less conducting targets than previously possibly with LFEMI. This makes possible the detection and identification of some IED constituent parts such as carbon rods and wires which we will discuss in the following sections.

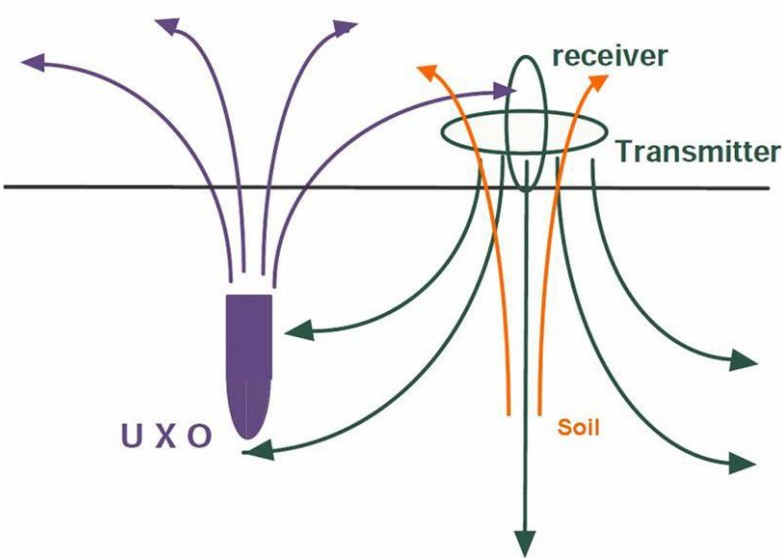

**Figure 2.** An EMI sensor field impinges on a UXO, which tends to respond most strongly along its axis, producing a response that strikes the receiver at an angle relative to the vertical. The secondary field is comprised of the response from the target and from the soil.

## 3. Materials and Methods

The HFEMI sensor consists of a frequency generator and analog-to-digital converters, sometimes combined with amplification on the receive channels. For the results presented in this paper, we used a National Instruments Virtualbench device for the function generator as well as the analog-to-digital converters via the oscilloscope channels [26]. For amplification, we used stock amplifiers applicable in this frequency range such as the Texas Instruments THS4631DDA. The data acquisition procedure requires three steps, acquiring a background acquisition, acquiring data from a piece of ferrite, and finally acquiring data from the target under consideration [13]. We measure the currents from the function generator for later data normalization as well as the voltage from the received coil after amplification. We programmed the Virtualbench in LabVIEW to sequentially cycle through logarithmically spaced frequency ranges typically from 500 Hz to 15 MHz. These data were sampled at a sampling rate which was at least 20 times the frequency of the transmitted signal, and typically 100 to 1000 full waveforms of data were recorded at each frequency. These time domain data were then used to extract the inphase and quadrature components of the received field via correlated deconvolution: multiplying the Rx data by a sine and cosine extracted from the Tx data and integrating. We then combined these in phase and quadrature components from the background, ferrite, and target acquisitions to both subtract additive noise including the effects in the data from the soil, and to divide out multiplicative noise [13]. At Fort AP Hill and for all the results in this paper, the background data was acquired with the sensor placed on the soil. As a result, soil effects were subtracted out of the subsequent target data. In this way, we were performing a change detection by subtracting the soil effects from the target data.

We have performed simulations on the effect of soil on the HFEMI received signal as well as the response of the targets presented in Section 4 [6,26,31,50] using the method auxiliary sources (MAS [51–55]) full wave numerical model. The MAS model solves boundary value problems numerically by representing the electromagnetic fields in each domain of the structure under investigation by a finite linear combination of analytical solutions to the relevant field equations, corresponding to sources situated at some distance away from the boundaries of each domain. The "auxiliary sources" producing these analytical solutions are chosen to be elementary dipoles/charges located on fictitious auxiliary surface(s), usually conforming to the actual surface(s) of the structure. In practice, we only require points on the auxiliary and actual surfaces, without resorting to the detailed mesh structures as required by other methods. The two auxiliary surfaces are set up inside and outside the scattering object. Specifically, the fields outside of the structure are considered to originate from set auxiliary magnetic charges placed inside the object, and the fields penetrating inside the object arise from a set of auxiliary magnetic dipoles placed outside the object. These fields constructed inside and outside of the object are required to obey the continuity of the tangential magnetic field components and the jump condition for the normal magnetic field components, at an array of selected points on the physical surface(s) of the structure. The results are matrix equation in which the amplitudes of auxiliary sources to be determined. Once the amplitude of auxiliary sources is found the solution is complete, the magnetic or electromagnetic field and related parameters can easily be computed throughout the interior and exterior domains. Thus, the MAS formulation offers several advantages: there is no longer need to integrate to produce the algebraic system. Field singularities at source locations need not be confronted directly since the auxiliary surfaces containing the sources are separated from the physical surface where conditions are applied. No mesh discretization of either the surfaces or volumes of interest is required, only location of points on the real and auxiliary surfaces.

Currents on wires can be excited by two predominant methods at these low frequencies. The physics of how these two different types of currents are excited aids in understanding what to expect from their resulting fields [20,32,33]. During electromagnetic induction and scattering, an external primary electromagnetic field causes currents in metallic objects. The total current in the target is then

$$\mathbf{J} = \sigma\mathbf{E} + \frac{\partial\mathbf{D}}{\partial t} \tag{1}$$

where $\sigma$ is the electrical conductivity, $\mathbf{E}$ is the electric field, $\mathbf{D}$ is the electric flux density, with $\sigma\mathbf{E}$ representing conduction currents and $\frac{\partial\mathbf{D}}{\partial t}$ displacement currents. The total electromagnetic force exerted on electrons in the metal, according to the Lorentz force equation is,

$$\mathbf{F} = e\left(\mathbf{E} + [\mathbf{v} \times \mathbf{B}]\right) \tag{2}$$

where $e$ is the electron charge, $\mathbf{B}$ the magnetic flux density, and $\mathbf{v}$ the velocity. At low frequencies, the electric field is small because in metals, conductivity and electron mobility are high. Consequently, the electric field force $\mathbf{F}_e = e\mathbf{E}$ is smaller than the magnetic field $\mathbf{F}_m = e(\mathbf{v} \times \mathbf{B})$ force. As a result, the magnetic $\mathbf{F}_m$ force, which is perpendicular to the velocity and magnetic field, pushes electrons to move from the interior volume of the metal towards its surface. In turn, the electrons form closed loop eddy currents, which satisfy $\nabla \cdot \mathbf{J} = 0$. The induced eddy currents produce a magnetic field, which opposes the changing flux inside the object. As the frequency increases, electrons are redistributed nearer the surface of the conductor in such a way as to make total electric $\mathbf{E}$ and magnetic $\mathbf{B}$ fields trend toward zero inside the conductor. At the same time, these moving charges on the surface of the conductor ensure the tangential $\mathbf{E}_t$ and normal $\mathbf{B}_n$ components of the electric and magnetic fields obey the proper surface boundary conditions. In addition, at high frequencies the magnitude of the electric

field force becomes larger than the magnitude of the magnetic field forces leading to surface currents based on the time rate of change of charge density as

$$\nabla \cdot \left( \mathbf{J} + \frac{\partial \mathbf{D}}{\partial t} \right) = \nabla \cdot \mathbf{J} + \frac{\partial \rho}{\partial t} \tag{3}$$

Given the proper orientation of the excitation electric field, these electrons on the wire then form surface (or linear) currents along the surface of the wire in the long direction, eventually forming a half wave dipole antenna given high enough frequency. These linear currents, as opposed to inductive currents that form in loops, produce fields with different shapes and geometrical spreading factors compared to the fields produced by inductive loop currents. Section 4.3 shows results of measuring the magnetic field near wires both in the lab and in the field over the HFEMI frequency range for both inductive and linear currents.

## 4. Results

The development of the HFEMI hardware can be found in more detail in other sources [26]. The measurement procedure for the HFEMI sensor is similar in concept to infrared emission spectroscopy (IRES) in that three measurements are required in the data acquisition process. A background is first acquired with no target present with the purpose of characterizing ambient noise sources. Next, a response from a substance with a known response, gray graphite in the case of IRES [56], ferrite in the case of HFEMI [13], is acquired. These two responses are then used to remove additive and multiplicative sources of noise, respectively, from the subsequent data.

The measurement procedure for all data reported in this paper was as follows:

1. Isolate the HFEMI instrument from known targets except for the ground (for field results)
2. Set LabVIEW program to program VirtualBench to cycle through *n* log-spaced frequencies (often *n* = 50 from 50 kHz to 15 MHz)
3. With these parameters, acquire 100 waveforms of data with sampling frequency at least 100*frequency as a background
4. Place a piece of ferrite near the sensor, acquire data again. Remove ferrite.
5. Place a target near the sensor (or move sensor over a subsurface target), acquire data again
6. Use the method described in [13] to extract the I and Q of the target
7. Repeat steps 5 and 6 for next target

### 4.1. Laboratory Results

Using this measurement procedure, we were able to acquire data such as those shown in Figure 3.

The data shown agree well with data acquired with low-frequency EMI sensors, displaying a quadrature peak and a high-frequency inductive limit. Even though the shotput (inferred conductivity $10^7$ S/m and permeability $\mu_r = 150$) was relatively close to the sensor, only 6 inches away to the nearest point on the sphere, these data agree well with MAS modeling. Figure 3b shows data acquired from a non-permeable carbon fiber plate with conductivity around 2500 S/m. The salient characteristics of the data curves represent a signature that is readily distinguishable between different shaped and conductivity targets in this frequency range just as the corresponding low-frequency EMI data curves represent signatures in the low-frequency range for EMI. We include in Figure 3b reference data from a commercially available frequency domain LFEMI system, the GEM-3 by Geophex, Ltd. (Raleigh, NC, USA). The GEM-3 scaled data and the HFEMI data both agree in shape with the MAS numerical model.

Several other lab grade measurements further illustrate the applicability and the capability of the HFEMI sensor to identify different targets. Figure 4 shows four more results from the HFEMI sensor over relevant targets in the laboratory. Figure 4a shows data acquired over two different diameters and thicknesses of carbon fiber pipe. Please note that the quadrature peak and magnitude of the response are easily distinguishable to the point that these targets could be identified by a single broadband

HFEMI measurement. Figure 4b shows data acquired over a tire pressure plate which includes an aluminum plate with a wire attached surrounded by rubber without steel radials. While this pressure plate displays some properties of a signature even at frequencies under 100 kHz, a combined low and high frequency measurement of this pressure plate would better characterize it and lead to a more trustworthy identification. Figure 4c shows data acquired over a carbon rod such as would be found in a D cell battery. These carbon rods have conductivity in the low thousands of Siemens per meter and are used for the express purpose of not being able to be detected with low-frequency metal detectors. This carbon rod exhibits a signature in the HFEMI band above 100 kHz, but shows essentially no response at frequencies under 100 kHz. Figure 4d shows data acquired over a depleted uranium penetrator. These depleted uranium penetrators are found on military ranges such as at Yuma Proving Grounds and have been used in conflicts in Europe and the Middle East. Depleted uranium has slightly lower conductivity than metals such as aluminum, and this reduced conductivity can be recognized in the data as a quadrature peak shifted to higher frequencies than the peak from other metals given the same size and shape target. Figure 4e shows data acquired over a lower-metal-content landmine, the VS-50 landmine. This landmine has a quarter-sized metallic plate plunge which makes it also easier to identify at lower frequencies, though HFEMI frequencies help with ID in this case. Finally, Figure 4f shows that even very thin layers of metal such as the gold foil on a SIM card are readily detectable with the HFEMI sensor with this SIM card being detectable at 3 to 4 inches suitable for an application in a wand, for example.

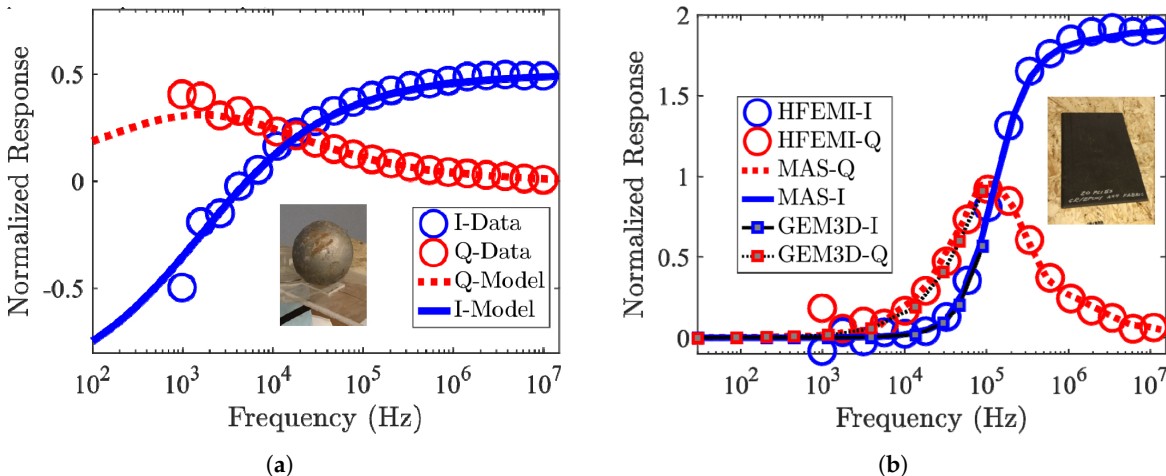

(**a**)  (**b**)

**Figure 3.** Examples of HFEMI data acquired over targets in a laboratory setting compared to MAS model predictions. (**a**) Represents HFEMI data acquired from a 4 inch shotput; while (**b**) shows HFEMI and LFEMI (Geophex GEM-3) data acquired from a 4 mm thick CF plate.

Improvised explosive devices continue to be the principal asymmetric threat against dismounted troops in theater. State of the art handheld instruments meant to detect buried threats consist of both low-frequency EMI sensors and GPR sensors. While EMI can detect metals and GPR detects dielectric or conductive heterogeneities, both fail to detect and classify some types of IED constituents. Figure 5 shows some typical mock IEDs similar to those found in conflict zones: zigzag wire pressure plates, wooden plungers, tire pressure plates, and carbon rods. These Mock IED were provided by the Office of Naval Research, and similar stand in IED targets were provided by the Night Vision and Electronic Sensors Directorate at their Fort AP Hill facility mentioned below.

Not pictured is another common type of IED and that is the ammonium nitrate fuel oil (ANFO) IED which is usually accompanied by a wire. Current handheld sensors such as the US Army's Handheld Standoff Mine Detection System (HSTAMIDS) and the Anglo-German Vallon–Cobham VMR3 Minehound do not use the frequency spectrum between 50 kHz and 20 MHz (the HFEMI frequency regime), and therefore lack the capability to detect and classify these IED constituents.

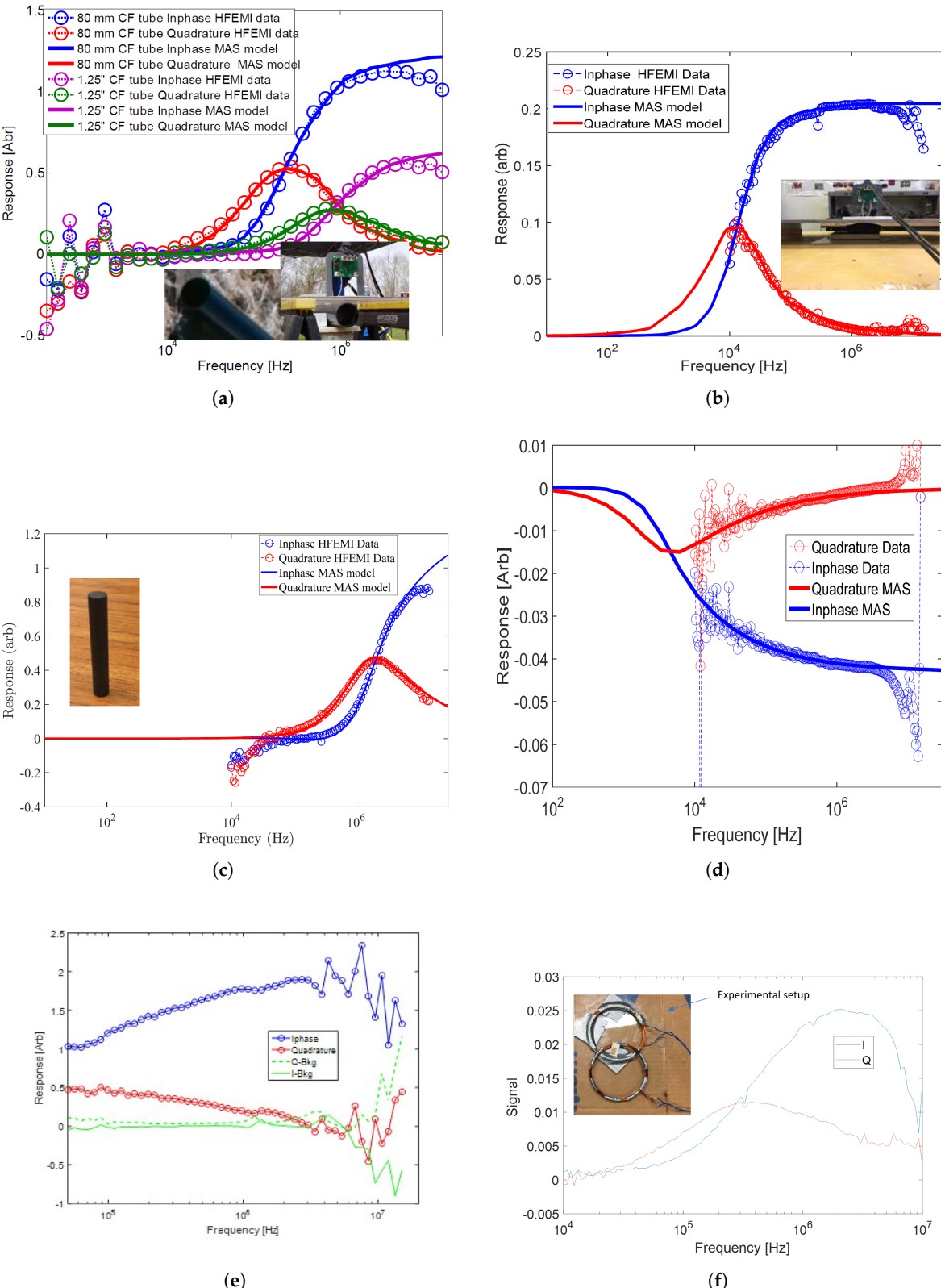

**Figure 4.** HFEMI data from carbon (**a**) fiber pipes; (**b**) a tire pressure plate; (**c**) a carbon fiber rod; (**d**) a depleted uranium penetrator; (**e**) a VS-50 landmine; and (**f**) a SIM card showing different signatures for each target.

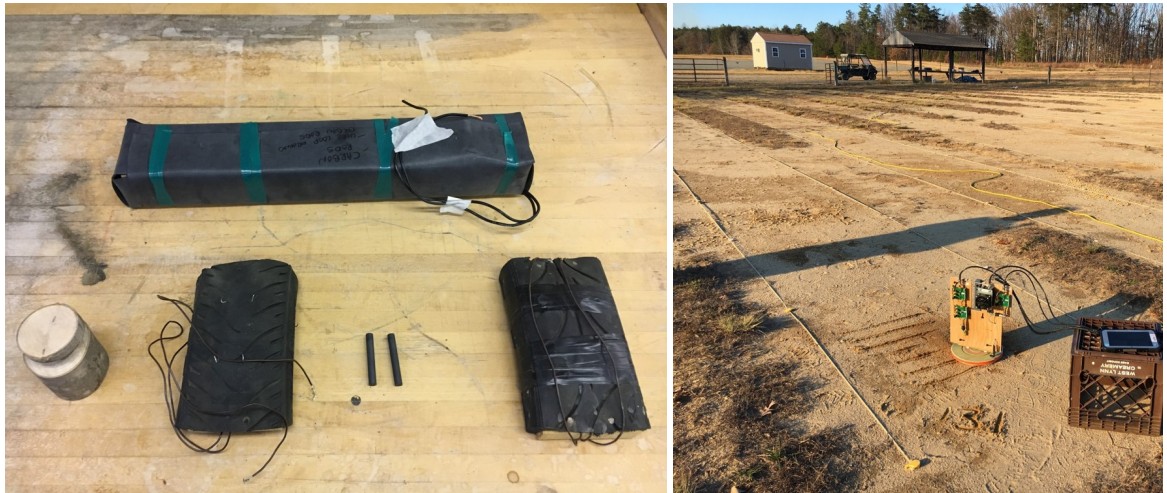

**Figure 5.** (**Left**) Mock IEDs designed to typify those found in theater; (**Right**) The Geophex type sensor deployed at Fort AP Hill IED test lanes.

## 4.2. Fort AP Hill Data

Fort AP Hill, under the direction of the Night Vision and Electronic Sensors Directorate, maintains IED test lanes containing IED constituent parts buried at different depths. Fort AP Hill is in Caroline County VA, and soils at the test lanes were mostly comprised of a sandy loam with low to moderate magnetic susceptibility native to the area [57]. We took our HFEMI instruments there as a test of sensor's capabilities outside of a laboratory environment (see Figure 5 (right)). We tested the sensor over various targets such as a zigzag wire carbon rod pressure plate IED buried in local Virginia soils. Soil does respond to the HFEMI sensor, although our background subtraction scheme tends to mitigate soil effects, which may be considerable. We intend to address the response of soils at HFEMI frequencies in future work. Figure 6 shows the results in which the contribution from the carbon rod with a quadrature relaxation peak near one MHz is clearly identifiable.

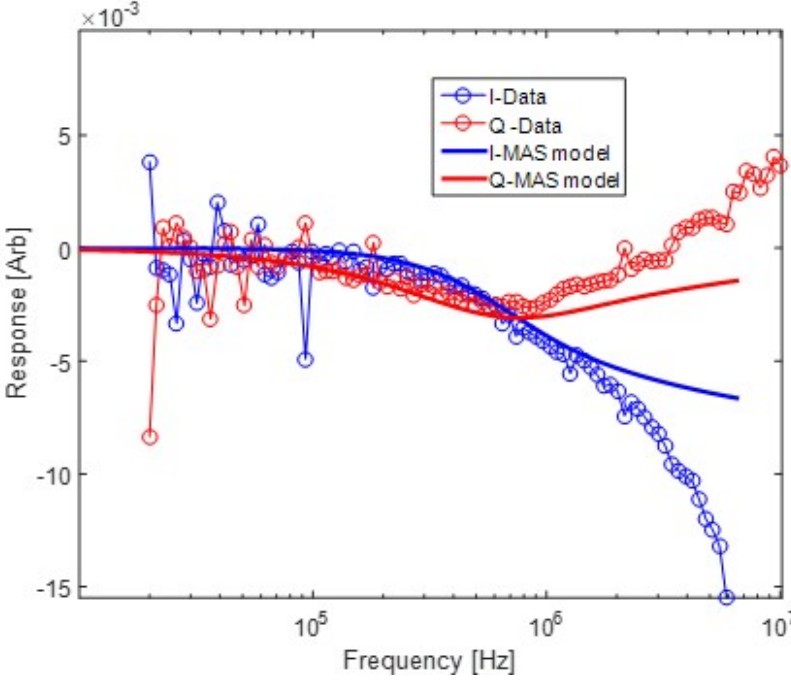

**Figure 6.** HFEMI response from carbon rod and zigzag wire in a mock IED.

Also shown are results from the MAS model of the carbon rods but not the wires in the pressure plate. The deviation of the data from the model at higher frequencies reflects the presence of the wires in addition to the carbon rods, while the apparent shift of the quadrature peak to lower frequencies is due to the increasing influence of the zigzag wire in the combination pressure plate IED.

ANFO IEDs are essentially conductivity voids in the soil. As such, normally these would be undetectable by low-frequency EMI sensors, but HFEMI can detect the absence of slightly conductive soils, i.e., a response from the conductivity void in the soil. Figure 7 shows this response as we moved the sensor progressively along the axis of the ANFO IED.

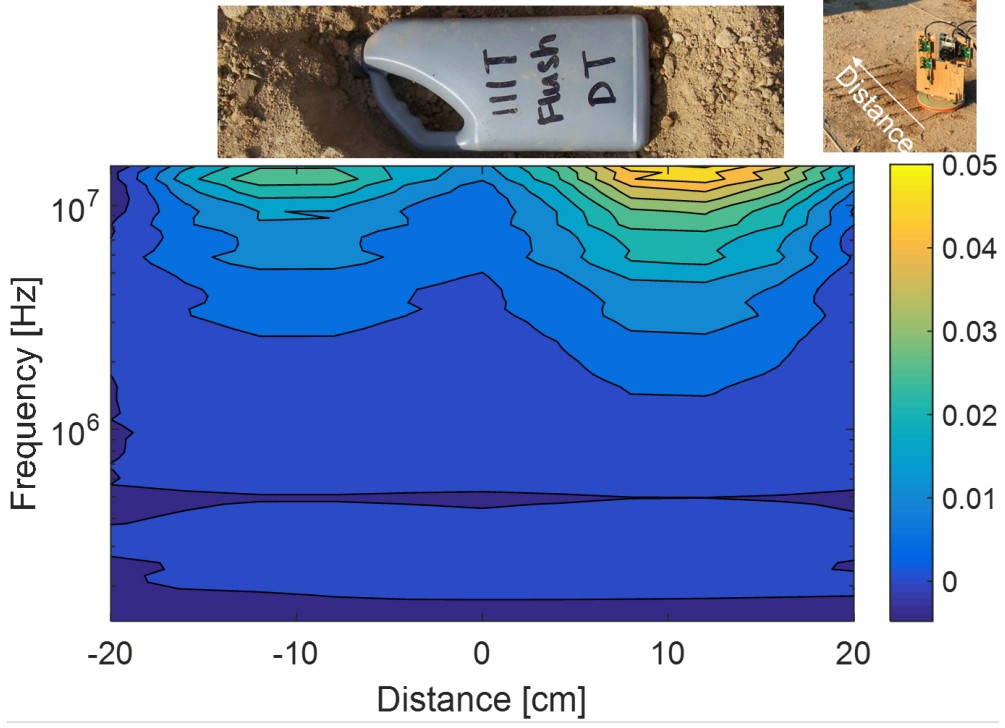

**Figure 7.** HFEMI inphase response at 10 MHz in a line over a jug of ANFO, essentially a conductivity void in the soil.

As previously mentioned, wires, whether short (<1 m) or longer command wires (>1 m) are not thick enough to sustain sizable eddy currents and therefore are largely undetectable by low-frequency EMI as well as GPR (c.f. Figure 10). However, Figure 8 shows that both shallow and deeper wires are clearly detectable with HFEMI. We have not performed a systematic test of the range at which the short wires can be detected with the HFEMI sensor. In a similar application at different frequencies however it has been shown that long metallic urban infrastructure can be detected using similar principles at 10 to 15 m depth [58,59].

The range at which these short wires are detectable is greater than the range at which carbon rods, for example, are detectable due to the linear currents set up on the wire that broadcast a magnetic field with a less severe geometrical decay than the more compact carbon rod targets [32,33]. The detection range improves near $n\lambda/2$ resonances where it approaches a multiple of a half wave dipole antenna.

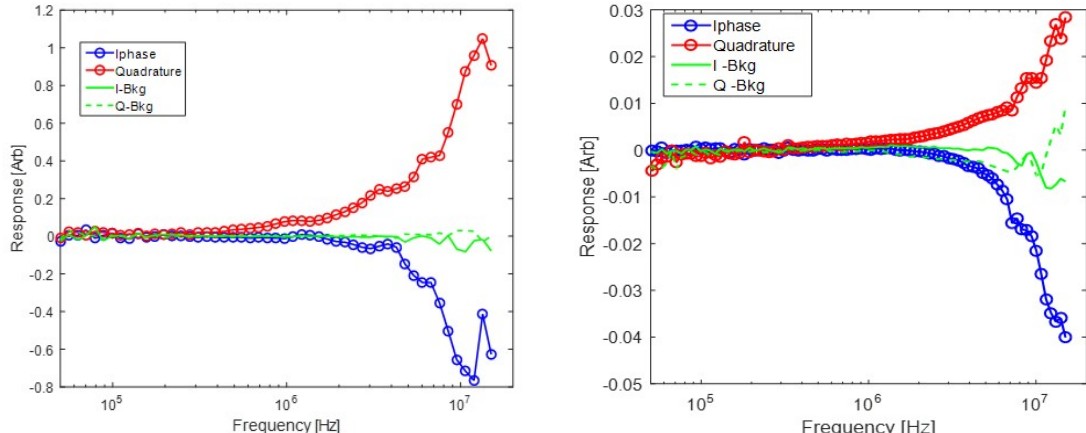

**Figure 8.** (**Left**) HFEMI data acquired over a shallow wire (≈1″ deep); (**right**) response from similar wire buried at 4″ depth.

### 4.3. Wire Detection at HFEMI Frequencies

Detecting thin wires with diameters on the order of 1 mm or less have proved very difficult for currently available sensors. As shown below, low-frequency EMI sensors cannot create large enough inductive currents to measure the secondary response due to their low frequencies. GPR sensors will detect reflections from wires, but the signature is mixed in with reflections from other inhomogeneities, especially because wires in IEDs tend to be near the surface at the air soil boundary. The HFEMI sensor can detect wires with an inductive response and with a linear current response.

The changing nature of the current distributions in the wire as a function of frequency can be seen in Figure 9 which presents results from MAS simulations of currents inside a wire at three different frequencies. Shown here is the current magnitude in a cross section of a copper wire ($\sigma = 5.8$ MS/m) of 2 mm diameter and 10 m long. The wire is illuminated with a coplanar, 15 cm diameter circular loop of current 1 cm away from the wire. As the frequency increases from 5 kHz to 50 kHz and then 500 kHz, the current distribution moves from inductive volume currents forming closed loops to surface displacement currents traveling along the length of the wire. These surface currents radiate more readily, especially at frequencies high enough to form appreciable fractions of a wavelength over the length of the wire. These radiative fields can be sensed further away than can the magnetic fields due to quasistatic inductive currents.

Figure 10A shows both the modeled and measured inphase and quadrature responses of a 30 cm long 12 American wire gauge (AWG) wire over the frequency range from 10 kHz to 15 MHz.

The model fits the measured inphase and quadrature responses well between 10 kHz and 10 MHz. Because we used fewer turns in our Tx and Rx coils, data at low frequencies are noisier than with traditional EMI sensors. The center frequency of the relaxation peak in the quadrature component of this data from this particular wire fragment occurs at approximately 1.8 MHz. Figure 10B shows the experimental set-up for the measurement. Note the overlapping receiver and transmitter coil configuration as an alternative primary field bucking technique. This "double-O" configuration was chosen as an alternative to the figure-8 Rx coil method to maximize the response of the wire. In this case, the wire should be close to the edge of the Tx coil so that the electric field from the Tx coil will be the strongest and aligned with the wire under test.

The response for long wires, for example wires in tunnels or command wires, is different when compared to the case of the short wires. The response as indicated above follows a linear current radiating rather than an induced current. We tested a 26 m long wire to see whether we could cause a linear current along the wire, see Figure 11. A 26 m wavelength in free space corresponds to a frequency of about $3 \times 10^8 / 26 = 11.5$ MHz. Due to end effects and coupling of the wire with the metal in the room and rebar in the floor, the full wavelength instead fits on this 26 m wire at 7.5 MHz with

the half wave resonance occurring at 3.8 MHz. These resonances suggest that indeed, linear currents are causing a radiated electromagnetic field around the wire in addition to an EMI magnetic field which is swamped out by the much larger fields due to the linear currents.

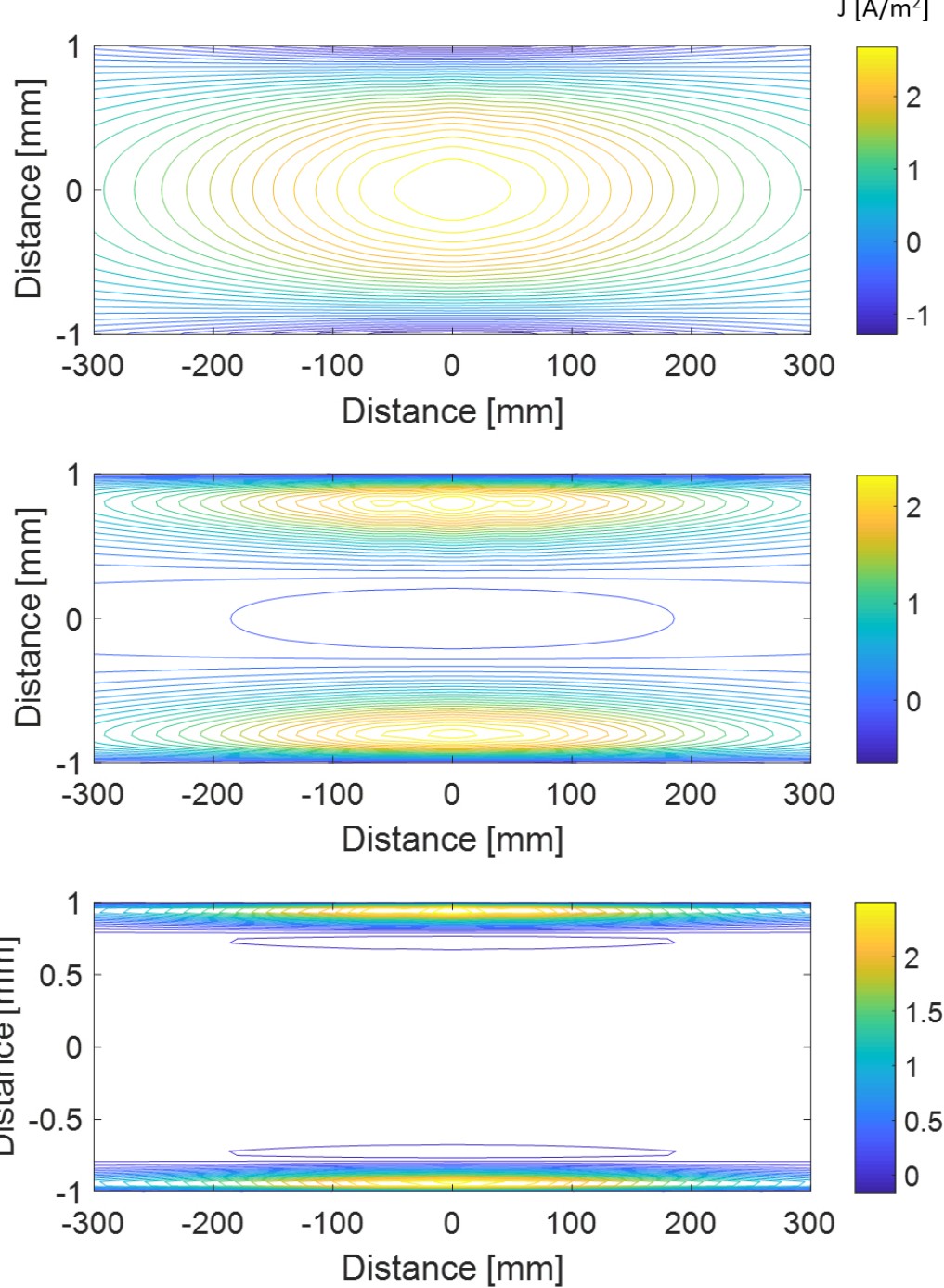

**Figure 9.** Current density inside a copper wire at 5 kHz (**upper**); 50 kHz (**middle**); and 500 kHz **Lower**).

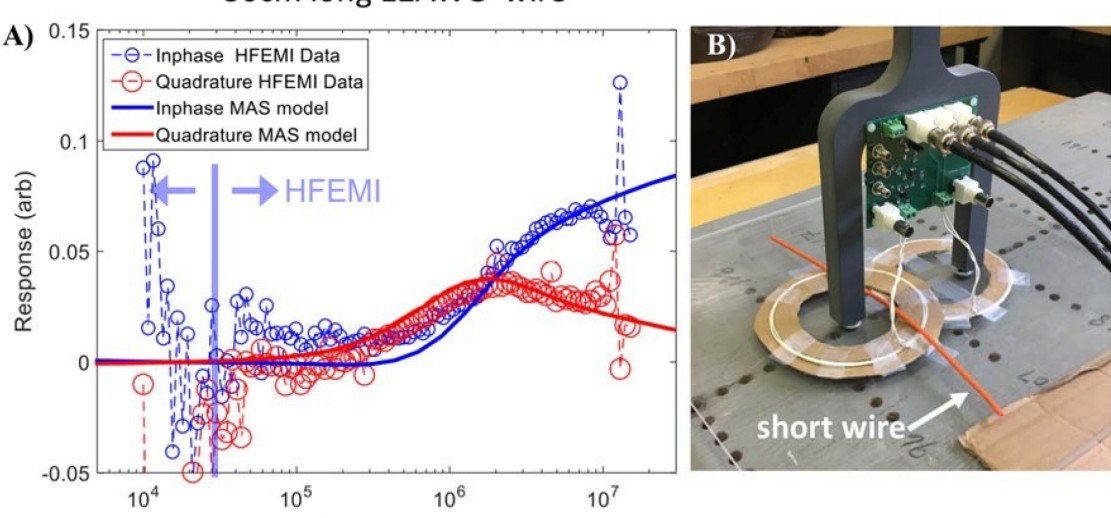

**Figure 10.** (**A**) HFEMI measurements (circles) and models (solid) for inphase (blue) and quadrature (red) response of a 30-cm, 12-gauge wire; (**B**) experimental set-up for the measurement displaying the HFEMI antenna configuration.

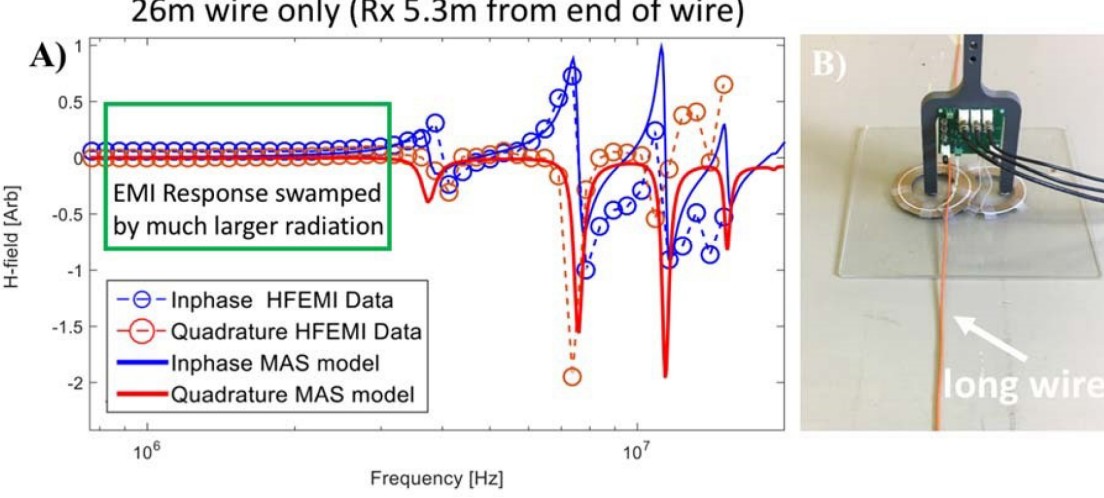

**Figure 11.** (**A**) HFEMI measurements (circles) and models (solid) for inphase (blue) and quadrature (red) response and model of a 26-m, 12-gauge wire, with the receiver positioned 5.3 m from the end of the wire; (**B**) experimental set-up for the measurement displaying the MFEMI antenna configuration.

## 5. Discussion

The HFEMI sensor represents a new way of interrogating near field anomalies. While prior sensors have investigated relaxation responses at frequencies under 100 kHz, and some ground penetrating radar have used frequencies down to 10 MHz for reflections and visual-based sensing, this sensor is the first to look at relaxation responses from discrete macrotargets in the frequency range of 100 kHz to 20 MHz. The advantages of interrogating discrete targets in this frequency range include retaining material information about the target such as conductivity, being able to account for ground effects and thus interrogate subsurface targets without disturbing the interstitial medium, and the ability to acquire a signature from targets which can then be used for ascertaining target parameters such as size, shape, location, conductivity, and dipole moment. However, with this higher frequency compared to traditional EMI, reduced coil length diminishes the SNR and limits the range of this version of the sensor. In addition, the HFEMI sensor is responsive to more of the environment,

such as the ground and saltwater/humans. While this can be a negative if one is only interested in subsurface discrete targets such as UXO and IEDs, this response to the environment may be a positive for applications regarding interrogating the permittivity of soils and the presence or properties of these weakly conducting media.

## 6. Conclusions

The high-frequency electromagnetic induction sensor constitutes a new way of looking at discrete targets in the frequency range of 50 kHz to 20 MHz. In this frequency range, less conducting targets as well as conventional small metal targets display inphase and quadrature signatures that can be used to detect and classify discrete targets that have proved undetectable by current sensors. At these frequencies above 50 kHz, but below typical ground penetrating radar frequencies of 20 MHz, less-conducting or small-profile targets such as carbon rods, carbon fiber ordnance, thin wires, low metallic landmines, pressure plates, and even conductivity voids exhibit HFEMI signatures. Magnetoquasistatic assumptions still apply in the near field at these frequencies when considering discrete targets, which facilitates conventional analysis methods such as those applied to low-frequency EMI. Linear currents excited on wires have less geometrical spreading and can therefore be detected from longer distances than discrete targets with secondary fields due primarily to inductive loop currents. Results from data acquired at the Fort AP Hill IED test lanes suggest that HFEMI could become a viable tool that could be added to existing low-frequency EMI instruments in order to better assess and classify threats in theater.

## 7. Patents

The HFEMI sensor has been patented with # 10,001,579.

**Author Contributions:** Initial HFEMI hypothesis and theoretical studies were performed by B.B. and F.S.; B.B. then built the first prototype HFEMI sensor and also subsequently contributed to project vision and management; G.J., B.B., and F.S. performed the data collection for this paper while F.S. performed the data analysis and modeled the results; M.P. designed and implemented the initial prototype hardware used in this project; while G.J. oversaw later hardware development; B.B. then wrote the manuscript with editing and suggestions from all coauthors.

**Funding:** This research was funding by the ERDC Environmental Quality and Installation UXO program as well as the Office of Naval Research (N000141612332).

**Acknowledgments:** The authors would like to thank Cory Shpil from the Night Vision Electronic Sensor Directorate, Countermine Division for his help and direction as well as Mel Soult at Fort AP Hill for his help acquiring data at the IED test lanes.

**Conflicts of Interest:** The authors declare no conflict of interest.

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
