# Peer review of "High-Frequency Electromagnetic Induction (HFEMI) Sensor Results from IED Constituent Parts"

_remotesensing, doi:10.3390/rs11202355_

Round 1
Reviewer 1 Report
This is an excellent paper.
It addresses a very important topic: Locating IEDs.
I believe that the authors should address the issue of FCC compatibility with these HF measurements. EMI systems in this frequency range need to be put into operational use as soon as possible in order to help locate IEDs. FCC compatibility will have to be addressed before any operational use. The authors may choose to address the FCC compatibility in a later paper, but I hope they can begin this work soon.
A few minor issues:
Misspelling on line 37 - classigying.
The caption in Figure 3 should include a brief description of a and b.
The caption in Figure 4 should include a brief description of a - f.
Lines 261-263 seem to be an incomplete sentence.
.........................................
I strongly recommend acceptance of this paper.
Reviewer 2 Report
This MS has potential to be a great paper. What's needed are following:
Must greater detail in methods and materials section. A reader skilled in the art, such as myself, should be able to build and duplicate your test results from this description. This is a critically important backbone to all scientific work, the research community must be able to duplicate and verify your results. Given this has potential to detect IED and land-mines, I can't think of research that has potential to truly benefit man-kind. But that can only do that if you explicitly enable reader to build the device with a very detailed description of your system and the testing protocols you followed to prove how well it works. considering you're using I/Q modulation processing, sufficient detail on the specifics on how you're creating, receiving the interrogation pulses as well as the specific digital signal processing filters you're using are also critically important to the performance and the reader's ability to duplicate this work. If you want to truly save lives, then be as specific as possible into how this thing works so that readers of this open publication can run with a working design. Soil specific issues cause havic with these types of sensors, so ommision of this information in design of the testing is a very important breech in protocol which gives one pause that the testing was very insufficient. Specifically needed, yet no mention, in materials and methods section was given to: soil type soil salinity soil moisture content soil iron content soil clay content and type of clay, %sand, % silt I couldn't find where you came up with the design for the mock IED. If someone is going to risk their life on this design, they need to be assured your mock IED is a conservative representative of a real IED that could likely explode and kill the user if this thing doesn't work... a lot of responsibility on the authors of this incredibly important work.
Summary, from write up I gather this likely works, but the details for us to be able to verify and duplicate this work is critically lacking and without this reviewer won't be able to accept for publication till it is. As to experimental protocol, there's not enough information to tell if it was sufficiently tested, so will have to await judgement on that. Would suggest in very least running test for following soil-types: in a sand, loam and clay for dry, low-saline near saturation (field-capacity) and high-saline near-saturation. As iron content is going to affect things, would also suggest a soil with low, normal and high iron content.
Reviewer 3 Report
A few typos or suggestions :
L7: promising
L34: such
L: ordnance
L91: end of the line, should be Δt and not its inverse
L145: Note that the quadrature peak in size of the response are
L159: rewrite 'then...target by than many metals given the same size and shape target'
L165: clarify your point 'similar form factor'
L178: of instead of on?
L180: replace 'given the' by 'due to their'
L181 : off of???
L184: aids
under eq 1 (no line numbers!) : displacement, replace 'B is' and 'v is' by 'B' and 'v the', small and because
Eq 3: '-' instead of '+'
L205: In general it is clear
L212: replace 'as might be associated with'
L213: above,
L226: megahertz
L229: replace 'in addition' by 'while'
L281 to 283: no verb in the sentence!
L284: targets,
Reviewer 4 Report
Please see attached file

Round 2
Reviewer 2 Report
In your response to my review, you mention you already covered the information I requested in earlier papers, yet when I pulled those papers, you're missing most of what I asked for. Specifically; more detail is critically missing on signal processing; you do a great job in your references of going over theory. What's lacking is specifics that pertain to transition from theory to a working device. You're missing electronics schematics and digital signal processing chains. Given importance of topic, transferring this knowledge is critically important as well as an important part of the scientific process that will allow your peers to duplicate and verify this technology works. Without which I cannot pass this MS onto publication.
Also requires more detail on your experimental protocol; you basically skip over materials and methods section completely and launch into results section, directly from background/theory, without providing ANY detail on HOW the experiment was conducted, so that the reader can duplicate your experiment and test results. The materials and methods section is not an optional section, it's a critically important backbone of the scientific method.
Given important of soil moisture in this type of work, you can't ignore it. You need to report what the moisture content was as well as soil-type (%silt, %sand, %clay) and for inductive type work as this, also need permeability of the soil.
